# Inclusion Complexes of β-Cyclodextrin with *Salvia officinalis* Bioactive Compounds and Their Antibacterial Activities

**DOI:** 10.3390/plants12132518

**Published:** 2023-06-30

**Authors:** Stamatia Christaki, Revekka Kelesidou, Vaia Pargana, Evangelia Tzimopoulou, Magdalini Hatzikamari, Ioannis Mourtzinos

**Affiliations:** Department of Food Science and Technology, School of Agriculture, Aristotle University of Thessaloniki, 541 24 Thessaloniki, Greece; stamchri@agro.auth.gr (S.C.); revekka.kel@gmail.com (R.K.); vaiaparg@agro.auth.gr (V.P.); tzimopou@cheng.auth.gr (E.T.); magdah@agro.auth.gr (M.H.)

**Keywords:** sage, inclusion complexes, β-cyclodextrin, FT-IR, antibacterial activity, phytochemistry

## Abstract

In the present study, the formation of molecular inclusion complexes of *Salvia officinalis* (sage) bioactive compounds with β-cyclodextrin (β-CD) was evaluated. Sage essential oil (SEO)/β-CD inclusion complexes (ICs) were prepared by co-precipitation at iso-molecular concentrations, and Fourier transform infrared spectroscopy (FT-IR) was applied for the confirmation of the ICs’ formation. Quantification of the SEO in the inclusion complexes was performed spectrophotometrically at 273 nm using an SEO standard curve. The SEO and its inclusion complexes were evaluated for their antimicrobial activity against *Escherichia coli, Staphylococcus aureus* and *Listeria monocytogenes*. The results showed that β-CD effectively formed inclusion complexes with SEO in satisfactory yields. The antimicrobial activity of the SEO in prepared complexes with β-CD was exhibited against *L. monocytogenes* and *S. aureus* and was proportional to their concentrations but was less pronounced.

## 1. Introduction

Cyclodextrins (CDs) are cyclic oligosaccharides whose structure in space, either as crystalline solids or as solutions, is ring-shaped, with their interior being non-polar and their exterior being polar [1]. Their molecular shape enables them to form inclusion complexes (ICs) with a wide variety of compounds which enter their hydrophobic cavity either partially or completely, replacing some high-energy water molecules from the interior [2]. In the food, cosmetics and pharmaceutical industries, CDs are used for the stabilization and protection of sensitive bioactive compounds (e.g., vitamins, fatty acids, pigments) from adverse environmental conditions such as temperature, radiation and pH [3]. Their property of masking unpleasant or intense odors and flavors (e.g., essential oils) is also noteworthy. In addition, through ICs’ formation using CDs, the water solubility of the guest molecules increases, thus hydrophobic compounds can be easily incorporated in aqueous media [4]. Some studies also focus on the encapsulation of bioactive substances in CDs for the development of novel antimicrobial agents [5]. Recently, green chemistry principles has turned the research interest towards both eco-friendly extraction techniques and alternative green solvents which are mainly based on water, are non-toxic and present potent extraction capacity. Therefore, the use of aqueous solutions of CDs has been included in the context of green extraction media [6]. In fact, it has been reported that CD solutions can be used as alternative solvents for phenolic compounds extraction from various materials [7].

Among CDs, β-CD is becoming the most widely used cyclodextrin as it appears to be compatible with a variety of molecules whose molecular weight ranges from 200–800 g/mol and its cost is relatively affordable [8]. β-CD is composed of seven glucose units, and as a crystalline solid it is characterized as essentially odorless and white/whitish, while its aqueous solution is clear and colorless. It has moderate solubility in water; it is easily soluble in warm water and slightly soluble in ethanol (Regulation (EE) No. 231/2012). β-CD is an approved food additive under the code E 459, as listed in Annex II and Annex III of Regulation (EC) No. 1333/2008.

In recent years, there has been an increasing trend in consumer preferences for more natural and “clean label” products, leading the food industry to the investigation and evaluation of natural compounds as potential antioxidant and antimicrobial agents [9]. In this respect, essential oils and extracts of aromatic and medicinal plants have attracted the interest of researchers with the aim of incorporating them into food products as natural additives [10] due to their potent antimicrobial and antioxidant activity [11,12]. The incorporation of essential oils and extracts in food products can be performed either directly (in bulk form or through encapsulation in different carriers prior to incorporation) or indirectly (in active packaging and edible coatings) [13].

*Salvia officinalis*, commonly known as sage or Dalmatian sage, is a species of the angiosperm genus *Salvia*, one of the largest groups in the Lamiaceae family. Sage is an important aromatic and medicinal plant of the Mediterranean region and a source of phenolic compounds (e.g., phenolic acids, flavonoids, tannins) and terpenoids that are characterized by high antioxidant capacity [14]. They can be found in all parts of the plant, prevailing in the aerial ones (i.e., leaves). More than 120 components have been recognized in the essential oil from the aerial parts of *S. officinalis*, such as α- and β- thujone, camphor, borneol, α- and β- caryophyllene, 1,8-cineol and others, which are responsible for its robust, piquant and herbaceous scent with musty nuances [15,16]. The antimicrobial activity of sage essential oil (SEO) is attributed mainly to the presence of thujone, camphor and 1,8-cineole, monoterpenes with antibacterial and antifungal potential. The essential oil of *S. officinalis* has been shown to inhibit strains of *Salmonella enteritidis, Bacillus cereus, Bacillus subtilis, Candida albicans, Staphylococcus aureus, Shigella sonnei* and *Aspergillus niger* [14]. However, its activity against *E. coli* is not particularly strong according to Longaray Delamare et al. [17]. The incorporation of SEO in beef, combined with the ideal storage temperature, led to a bacteriostatic effect against *L. monocytogenes* [18].

Generally, SEO has been studied in the literature as a potential antibacterial and antioxidant agent in food preservation [9,18,19,20,21,22], especially in meat products [23,24]. However, the use of β-CD for the encapsulation and stabilization of SEO has been reported only by Tian et al. [25] and Nait Bachir et al. [26]. In the former study, it was demonstrated that the inclusion interactions of β-CD with *Salvia sclarea* (clary sage) essential oil were far stronger compared to other β-CD derivatives for the formation of molecular ICs and β-CD should therefore be preferred over other derivatives [25]. The same study also reported the protection of essential oil compounds inside the cavities of β-CD. Regarding the encapsulation of SEO in other delivery systems, the literature is limited compared to other essential oils of the Lamiaceae family [21,27,28,29,30]. Therefore, the purpose of the present study was to investigate the formation of molecular ICs between β-CD and SEO and evaluate their antimicrobial activity, aiming to facilitate their further application in the food, pharmaceutical and cosmetics industries as potential preservative agents.

## 2. Results

### 2.1. Encapsulation Efficiency

In the present study, for the formation of ICs, 5000 mg of β-CD (powder form) and 670 mg of SEO (iso-molecular ratio) were used. The encapsulated SEO in the β-CD was spectrophotometrically quantified and determined at 263.62 ± 8.35 mg, compared to the initially added 670 mg, thus resulting in an encapsulation efficiency of 39.35 ± 1.25%. The efficiency of the formation process of the ICs, also referred to as recovery, was determined at 85.30 ± 0.44%. Regarding the amount of pure SEO “loaded” in the ICs per weight unit, it was determined at 5.45 ± 0.16% since the total weight of ICs at the end of the formation process was 4837 mg.

### 2.2. FT-IR

In Figure 1, the FT-IR spectra of SEO, β-CD and ICs are presented. In the SEO spectrum, the first peak is observed at 3463 cm^−1^, followed by peaks at 2958, 2929 and 2872 cm^−1^. In addition, a strong sharp peak appears at 1741 cm^−1^, followed by smaller ones at 1645, 1160 and 1109 cm^−1^. In the FT-IR spectrum of β-CD, there is a strong and broad absorption band at 3309 cm^−1^, while a characteristic peak is also observed at 2922 cm^−1^. A small peak appears at 1642 cm^−1^, followed by peaks at 1152, 1077 and 1021 cm^−1^. Finally, peaks between 1000 and 700 cm^−1^ appear at 889, 852 and 754 cm^−1^. In the FT-IR spectrum of the inclusion complex, the absorption band of the β-CD at 3309 cm^−1^ appears slightly more intense and shifted from 3309 to 3304 cm^−1^. Moreover, the peaks at 3463, 2958, 2929 and 2872 cm^−1^ from the SEO spectrum are not present. The strong sharp peak at 1741 cm^−1^ of the SEO is also present in the spectrum of the inclusion complex, but its intensity is clearly reduced and is shifted towards 1736 cm^−1^. Finally, the peaks between 1454 and 1382 cm^−1^ are not detected in the inclusion complex spectrum. The above changes qualitatively confirm the encapsulation of the SEO in the cavity of β-CD and the formation of molecular ICs. 

### 2.3. Antibacterial Activity

#### 2.3.1. Sage Essential Oil

The results of the antibacterial activity examined with the agar well diffusion method regarding SEO are presented in Table 1. Since the sunflower oil, used as diluent, presented no growth inhibition against the studied microorganisms when tested alone (control), the results were attributed to the effect of SEO. The non-diluted (pure) SEO resulted in the highest antimicrobial activity against the Gram-positive bacteria *S. aureus* and *L. monocytogenes*, while no inhibitory effect was exhibited against Gram-negative bacterium *E. coli*. Specifically, the antibacterial activity of SEO was significantly higher in *S. aureus* compared to *L. monocytogenes* (*p* < 0.05). When SEO was diluted in sunflower oil (1:1), only *S. aureus* was affected but to a smaller extent compared to the un-diluted SEO, while dilution 1:3 of SEO in sunflower oil had no antibacterial activity.

#### 2.3.2. SEO/β-CD Inclusion Complexes

In Table 1, the results regarding the antibacterial activity of ICs are also presented. Based on the calculation of the loading capacity (% LC) (Section 2.1), the actual concentration of SEO in ICs was calculated. Based on the results, SEO encapsulated in the form of ICs with β-CD demonstrated antibacterial activity in much lower concentrations than pure SEO against both Gram-positive bacteria *S. aureus* and *L. monocytogenes,* while no inhibition occurred against Gram-negative bacterium *E. coli.* Similar to the previous results for pure SEO, the inhibitory effect was significantly stronger (*p* < 0.05) against *S. aureus* compared to *L. monocytogenes*. Despite the differences in the activity for the two microorganisms, the ICs’ antibacterial activity was proportional to their tested concentrations. Since no antimicrobial activity was observed in the case of β-CD solution alone, the inhibitory activity of the ICs was attributed to the encapsulated SEO that was released from the complexes.

## 3. Discussion

### 3.1. Encapsulation Efficiency

For the effective formation of molecular inclusion complexes, it should be ensured that both the shape and size of the compound of interest are suitable for the respective cyclodextrin. The stereochemistry, and subsequently the polarity, of the two molecules therefore play a crucial role in the subsequent strength of the created bond [31]. Since the bond is formed between the hydrophobic parts, hydrophobic molecules, such as essential oil constituents, are most favorable regarding an increase in their water solubility [31]. For the overall quantitative evaluation of the ICs’ formation, a series of equations have been developed over the years. Among them, encapsulation efficiency (EE%), expressed as the percentage of active compound successfully entrapped in the complex, is an important parameter regarding the ICs’ characterization. Kotronia et al. [32] studied the encapsulation of oregano essential oil in β-CD. Following the same method of co-precipitation and an oil: β-CD ratio of 1:1 M, the EE% was determined at 22.60%, significantly lower than that of the present study (39.35 ± 1.25%). In contrast, in a study by Anaya-Castro et al. [33], the EE% values ranged from 46.99 to 77.57% for the various oregano essential oil: β-CD ratios. Halahlah et al. [34], who studied the encapsulation of rosemary essential oil in β-CD by co-precipitation, found an EE% of 36%, which is a satisfactory percentage and is quite close to that of the SEO in the present study. The difference in the EE% among studies could probably be attributed to the different plant species tested, the different interactions and the affinity of the essential oil constituents with β-CD. Unfortunately, no direct effectiveness comparison can be made regarding SEO encapsulation in β-CD since there is no published research on the topic. However, the EE% of the present study is almost double that of Kotronia et al. [32] and is almost equal to that of Halahlah et al. [34], so encapsulation could be considered effective. Regarding the amount of SEO “loaded” per weight unit of ICs (5.45 ± 0.16%), the results are comparable to those of Anaya-Castro et al. [33] for Mexican oregano essential oil, which ranged from 2.29 to 7.86%. The difference in the values between the two essential oils could be attributed to their main compounds, α-thujone for sage and carvacrol for oregano. Due to their different molecular weight, structure and linearity [35], it seems that oregano essential oil, in appropriate proportions, interacts more favorably with β-CD than sage. The process efficiency (PE%) or recovery (R%) of ICs is a parameter that refers to the efficiency of the selected method (e.g., co-precipitation) for the complexes’ production based on the recovered amount (weight) of ICs and the initial amount (weight) of the individual components. In the case of the present study, the PE% was 85.30 ± 0.44%, which indicates a satisfactory result, compared to a similar study by Anaya-Castro et al. [33] where PE% was calculated at 93.50%.

### 3.2. FT-IR

Fourier transform infrared spectroscopy (FT-IR) finds wide application in the study of the structure of inclusion complexes that are mainly in the solid state [36], but it can also be applied to aqueous solutions [37]. It is considered one of the most suitable methods to study host–guest molecule interactions at a molecular level through the detection of changes in the vibrational spectrum of complexed molecules compared to free ones [36]. Therefore, FT-IR is applied in the case of cyclodextrin inclusion complexes and offers a qualitative characterization of the complexes. The obtained spectra are evaluated to confirm successful encapsulation [5].

In the SEO spectrum, the first peak at 3463 cm^−1^ is due to the presence of O-H group stretching vibrations, while at 2958, 2929 and 2872 cm^−1^ the peaks are a result of the symmetric and asymmetric stretching vibrations of the C-H groups. The strong sharp peak at 1741 cm^−1^ is characteristic of essential oils due to stretching vibrations of the C=O group, which is present in cyclic ketones with a ring-forming tendency, such as camphor and thujone (SEO compounds). The stretching vibrations of the C=C group are characteristic of the double bond found in terpenes, such as β-myrcene and camphene, and the corresponding peak was detected at 1645 cm^−1^. Additionally, bending vibrations of the CH_2_ group were detectable at 1454 cm^−1^, whereas the peaks at 1160 and 1109 cm^−1^ indicate the presence of terpenes possessing tertiary and secondary alcohols. Similar results were reported by other studies as well [38,39].

In the FT-IR spectrum of β-CD, there is a characteristic strong and broad absorption band at 3309 cm^−1^ due to the symmetric and asymmetric stretching vibrations of the hydroxyl groups (-OH) of β-CD, while at 2922 cm^−1^ a characteristic peak is observed due to the stretching vibrations of the C-H groups. The bending vibrations of the H-O-H groups are responsible for the appearance of the peak at 1642 cm^−1^, while the symmetric and asymmetric stretching vibrations of the C-O-C groups create the peaks at 1152, 1077 and 1021 cm^−1^. Finally, peaks between 1000 and 700 cm^−1^ appear at 889, 852 and 754 cm^−1^ due to the backbone vibrations of the C-H and C-C bonds of the glycopyranose ring of β-CD, as reported by Ren et al. [40] and Lin et al. [41].

In the FT-IR spectrum of the inclusion complexes, peak shifts, decreases in intensity and overlapping of some peaks are observed, compared to the spectra of the individual compounds [42]. Due to the replacement of water molecules in the β-CD cavity with SEO molecules, the absorption band related to the symmetric and asymmetric stretching vibrations of the hydroxyl groups (-OH) appears slightly more intense and is slightly shifted from 3309 to 3304 cm^−1^. Moreover, the peaks at 3463, 2958, 2929 and 2872 cm^−1^ from the SEO spectrum are not present, indicating that they were overlapped by the β-CD peaks. The strong sharp peak at 1741 cm^−1^, which is characteristic of the SEO (stretching vibrations of the C=O group), is also present in the spectrum of the inclusion complex, but its intensity is clearly reduced and is shifted towards 1736 cm^−1^. Finally, the peaks between 1454 and 1382 cm^−1^, which are due to bending vibrations of C-H bonds, are not detected in the inclusion complex spectrum. The above changes qualitatively confirm the encapsulation of the SEO in the cavity of β-CD and the formation of molecular ICs. Corresponding studies have reported that alterations in the guest molecule characteristic bands, such as the disappearance or broadening of some peaks as well as the fluctuations in the intensity of the peaks, are the result of limited stretching vibrations of the guest molecule caused by its encapsulation in the β-CD cavity [41,43,44].

### 3.3. Antibacterial Activity

#### 3.3.1. Sage Essential Oil

In the present study, SEO presented antibacterial activity only against Gram-positive bacteria *S. aureus* and *L. monocytogenes* but it was ineffective against Gram-negative bacterium *E. coli*. Generally, Gram-negative bacteria are more resistant in the presence of antimicrobial agents due to their outer membrane consisting of a layer of lipopolysaccharides which can limit the diffusion rate of hydrophobic compounds (e.g., essential oils) inside the bacterial cell [45]. The ineffectiveness of SEO against *E. coli* has also been reported in the literature [17]. For the Gram-positive bacteria, *L. monocytogenes* showed higher resistance than *S. aureus* in all tested concentrations. Higher antimicrobial activity was associated with higher 1,8-cineole concentrations in the SEO compared to monoterpene hydrocarbons. In addition, *L. monocytogenes* cell walls contain stable lipoteichoic acids. These amphipathic polymers are similar in function and structure to the lipopolysaccharides of Gram-negative bacteria. This structure, combined with the tolerance of *L. monocytogenes* to a wide range of conditions, justifies the weak antibacterial action of SEO at small concentrations [17].

#### 3.3.2. SEO/β-CD Inclusion Complexes

The encapsulation of SEO inside the cavity of β-CD appears to improve the antimicrobial properties of the essential oil in vitro. Although the ICs were not effective against *E. coli*, similarly to pure SEO, the antimicrobial activity against *S. aureus* and *L. monocytogenes* significantly increased. Since the main targets of antimicrobial action are on the surface of bacterial cell membranes and inside the cytoplasm, β-CD may have facilitated the diffusion of SEO constituents through the substrate and subsequently to the targeted areas by enhancing their water solubility [46,47]. This improved solubility and cell accessibility of the bioactive compounds enables the application of lower essential oil concentrations for achieving antibacterial activity [48]. Since limited data are available in the literature regarding the antimicrobial activity of SEO and β-CD inclusion complexes, more research is needed in the field to draw a comprehensive conclusion. Higher IC concentrations should probably be tested against Gram-negative bacteria such as *E. coli*, while it would be interesting to also investigate the antifungal activity of such ICs containing SEO to broaden their applications as antimicrobial agents.

## 4. Materials and Methods

### 4.1. Materials

Sage essential oil (α-thujone 31.25%, cineole 9.86%, camphor 9.62%, β-caryophyllene 6.89%, α-caryophyllene 6.33%) was supplied from Physis Ingredients (Serres, Greece). β-CD (98.0%) was supplied from TCI (Tokyo, Japan). Sunflower oil was purchased from the local market. Brain Heart Infusion (BHI) Agar (CM1136B) and Brain Heart Infusion Broth (CM1135) were purchased from OXOID LIMITED, Basingstoke, England. The microorganisms used for the assessment of the antibacterial activity were *Staphylococcus aureus* ATCC 25923, (American Type Culture Collection, Manassas, VA 20110 USA), *Escherichia coli* ATCC 25922 and *Listeria monocytogenes* DSM 15675 (DSMZ-German Collection of Microorganisms and Cell Cultures GmbH, Braunschweig, Germany). Sodium chloride (NaCl) and hydrophilic filters with a 0.22 μm pore size (Q-max, hydrophilic PTFE) were purchased from Merck, Darmstadt, Germany and Frisinette, Knebel, Denmark, respectively.

### 4.2. Inclusion Complexes’ (ICs) Formation Using SEO with β-CD

The ICs were prepared using the method of iso-molecular co-precipitation as it is considered more suitable for water-insoluble compounds [49]. For mole calculation in the case of essential oils, it is hypothesized that the essential oil consists entirely of the major compound (in this case, α-thujone) [46].

To a water: ethanol solution (2:1 *v*/*v*), an amount of β-CD was added to a final concentration of 10% *w*/*v*. After complete dissolution of β-CD (clear solution) through magnetic stirring at 50 °C, the heating was stopped and an iso-molecular quantity of SEO was added dropwise to the solution. The magnetic stirring continued for 3 h at ambient temperature (25 °C) for the formation of ICs. Then, the mixture was stored in the refrigerator (4 °C) for 24 h for crystallization and precipitation of the ICs. The mixture was then vacuum-filtered, and the complexes were collected in paste form. The paste was washed twice with ethanol to remove SEO traces from the surface of β-CD and then dried in an oven at 35 °C for 24–48 h until reaching a constant weight. The ICs were collected in powder form. The experiments were performed in triplicate.

### 4.3. Encapsulation Efficiency

For the determination of encapsulation efficiency, quantification of SEO encapsulated in β-CD was performed spectrophotometrically based on an SEO standard curve (0.2 to 2 μL/mL) at 273 nm (Equation (1)).
y = 0.4337x + 0.004, R^2^ = 0.9997(1)

ICs in powder form were mixed with ethanol (1:5 *w*/*v*) in glass vials and immersed in an ultrasonic bath (Heated Ultrasonic baths “Ultrasons-H”, J.P. Selecta, Barcelona, Spain) for 30 min to disrupt the ICs and release the encapsulated SEO. The obtained mixture was centrifuged at 14480 rcf for 5 min (Hettich Universal 32, DJB Labcare, Buckinghamshire, England) and then filtered through hydrophilic filters (0.22 μm pore size) to remove any β-CD residues. The filtered solution was appropriately diluted in ethanol and the absorbance was measured at 273 nm. The results were expressed as mg SEO/mg ICs and the Equations (2)–(4) were used to evaluate the process efficiency/recovery (PE%), the encapsulation efficiency (EE%) and loading capacity/drug loading (LC%), respectively [33].
(2)$$\mathrm{PE}\;(\%)=\frac{\mathrm{total}\;\mathrm{weight}\;\mathrm{of}\;\mathrm{ICs}\;\left(\mathrm{mg}\right)\;}{\mathrm{initial}\;\mathrm{weight}\;\mathrm{of}\;\beta -\mathrm{CD}\;\left(\mathrm{mg}\right)+\;\mathrm{initial}\;\mathrm{weight}\;\mathrm{of}\;\mathrm{SEO}\;\left(\mathrm{mg}\right)}\times 100$$
(3)$$ΕΕ\;(\%)=\frac{\mathrm{weight}\;\mathrm{of}\;\mathrm{encapsulated}\;\mathrm{SEO}\;\left(\mathrm{mg}\right)}{\mathrm{initial}\;\mathrm{weight}\;\mathrm{of}\;\mathrm{SEO}\;\left(\mathrm{mg}\right)}\times 100$$
(4)$$\mathrm{LC}\;(\%)=\frac{\mathrm{weight}\;\mathrm{of}\;\mathrm{encapsulated}\;\mathrm{SEO}\;\left(\mathrm{mg}\right)}{\mathrm{total}\;\mathrm{weight}\;\mathrm{of}\;\mathrm{ICs}\;\left(\mathrm{mg}\right)\;}\times 100$$

### 4.4. Fourier Transform Infrared Spectroscopy (FT-IR)

For the qualitative evaluation of the ICs’ formation, Fourier transform infrared spectroscopy (FT-IR) analysis was performed using an IR 6700 spectrometer (Jasco, Essex, UK) equipped with a DLaTGS detector and a high-throughput Single-Reflection ATR with diamond crystal, accompanied by the Spectra Manager software (Jasco, Essex, UK). The spectra were accumulated at the transmittance mode at a resolution of 4 cm^−1^, and 32 scans were performed per sample, covering a wavelength range from 4000 to 400 cm^−1^. The analysis was performed in triplicate for each sample, and three spectra per sample were recorded and averaged before further processing. Changes in the peaks of the spectra and the appearance of new peaks in the spectra of the ICs (powder form) compared to the spectra of the individual compounds (SEO, β-CD) indicate the creation of new bonds and, by extension, the creation of molecular ICs between the essential oil and β-CD.

### 4.5. Antibacterial Activity

The antibacterial activity of SEO and its ICs in β-CD was tested against three foodborne bacteria: the Gram-negative *E. coli* ATCC 25922 and the Gram-positives *S. aureus* ATCC 25923 and *Listeria monocytogenes* DSM 15675, using the agar well diffusion method. The bacteria were activated by two successive subcultures in BHI broth and were incubated at 37 °C for 18–24 h. After two decimal dilutions in fresh saline solution (0.9% *w*/*v* NaCl), the inoculum was spread evenly on Petri dishes containing 20 mL pre-poured and solidified BHI agar, using a sterile cotton swab. After 30 min, three wells were formed in each Petri dish, using a sterile 1 mL tip cut at the end to achieve a wide bore of 4 mm in diameter; 30 μL of each sample was added separately into the wells.

The Petri dishes were left at 4 °C for 1 h for the diffusion of the sample into the substrate and then incubated at 37 °C for 18 h. After incubation, the diameters (in mm) of growth inhibition zones, including the diameter of the well, were measured. The samples tested were pure SEO, SEO diluted in sunflower oil (1:1 and 1:3) and ICs of SEO with β-CD. ICs in powder form were dissolved in water to appropriate concentrations, and based on LC%, the concentrations of pure SEO in the ICs correspond to 5, 4, 2, 1 and 0.5 mg of SEO/mL of water, respectively.

Sunflower oil, served as diluent, and aqueous solution of β-CD were also tested for possible antibacterial activity (control samples). All experiments were performed in triplicate, and the results are the mean value of inhibition zones ± SD.

### 4.6. Statistical Analysis

Statistical analysis was performed using SPSS software for Windows, version 27.0.1 (SPSS Inc./IBM Corp., Armonk, New York, NY, USA). The results were expressed as average values of three independent experiments ± standard deviation (SD), and one-way analysis of variance (one-way ANOVA), followed by Duncan’s post-hoc test, was performed to detect statistically significant differences. The level of statistical significance (*p*) was set at 5% (0.05).

## 5. Conclusions

In the present study, the formation of molecular ICs using SEO and β-CD was successful as the values of the encapsulation efficiency parameters were in line with similar studies in the literature. From the results concerning the antimicrobial activity, the ICs were much more effective against *S. aureus* and *L. monocytogenes*, proving that with less SEO concentration a satisfactory inhibitory effect could be reported. However, additional research based on both the same and other varieties of *Salvia officinalis* is needed for better evaluation of the antimicrobial activity of such ICs. In future research, ICs in powder form could be incorporated in various food systems prone to microbial spoilage and oxidation and studied as preservative agents aiming to extend their shelf life. Furthermore, their effect on the physicochemical and organoleptic characteristics of the final products could be evaluated.

## Figures and Tables

**Figure 1 plants-12-02518-f001:**
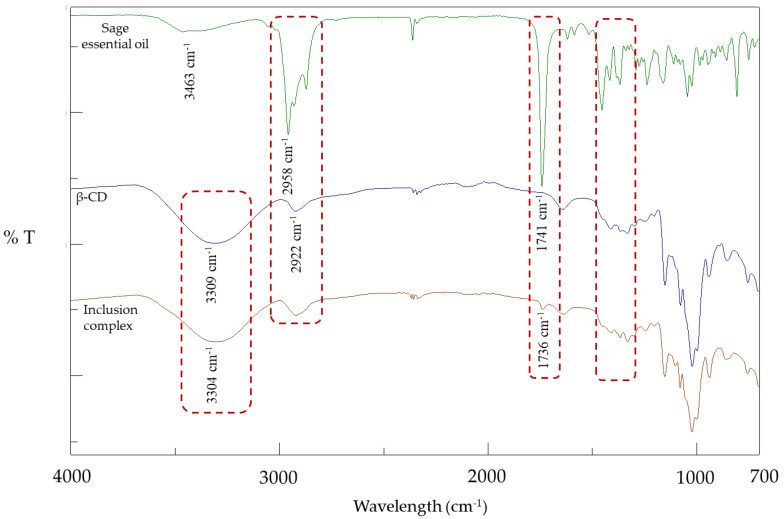
FT−IR spectra of sage essential oil, β-cyclodextrin (β-CD) and inclusion complex (the most characteristic peaks are noted); T: transmittance.

**Table 1 plants-12-02518-t001:** Antibacterial activities of pure SEO and inclusion complexes measured as inhibition zones in mm (mean values ± SD of three replicates).

	Antibacterial Activity ^1^		
	*L. monocytogenes* DSM 15675	*S. aureus* ATCC 25923	*E. coli* ATCC 25922
**Pure SEO**			
Un-diluted	23.3 ± 5.1	32.2 ± 6.7	ND ^3^
500 mg/mL	ND	7.2 ± 7.5	ND
250 mg/mL	ND	ND	ND

**Inclusion complexes**			
100 mg/mL (5.0 mg/mL) ^2^	20.0 ± 0.7	24.5 ± 0.7	ND
80 mg/mL (4.0 mg/mL)	16.5 ± 0.7	19.5 ± 0.7	ND
40 mg/mL (2.0 mg/mL)	11.0 ± 1.4	16.5 ± 0.7	ND
20 mg/mL (1.0 mg/mL)	8.0 ± 0.0	9.5 ± 0.7	ND
10 mg/mL (0.5 mg/mL)	5.0 ± 1.4	7.5 ± 2.1	ND

^1^ Diameter of inhibition zone including the diameter of well (4 mm); ^2^ concentrations in the brackets are the actual concentrations of SEO based on LC%; ^3^ not detected.

## Data Availability

Not applicable.

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
