# Peer review of "Inclusion Complexes of β-Cyclodextrin with Salvia officinalis Bioactive Compounds and Their Antibacterial Activities"

_plants, 2023, doi:10.3390/plants12132518_

Round 1
Reviewer 1 Report
The authors studied the formation of molecular inclusion complexes of sage bioactive compounds with β-cyclodextrin (β-CD). This topic is novel and of interest as there is no relevant literature available.
The manuscript is well written, the experimental design is sound and the statistical analysis is suitable. The authors also discussed their results and compared them to the current literature offering interesting and novel insights.
I believe this manuscript will be of interest to both academics and industry.
For the reasons I above, I recommend the manuscript to be accepted for publication after the minor comments below are addressed.
Minor comments
· Line 20, replace less effective with ‘’less pronounced’’
· Line 102, table 1. Since the values presented in Table 1 are also included in the text above, then it is not necessary to keep table 1. The authors should consider deleting it.
· Table 2 presents the antibacterial activities of pure SEO and inclusion complexes measured as inhibition zones. The results are presented in cm, however, the results should be presented in mm to coincide with the published literature. Please make the unit conversion.
· Also for table 2, please clarify for the reader in the footnote what it means when there is no value presented (‘‘-‘’ no antimicrobial effect)
· Line 338, please add the corresponding number for each bacterial species.
· Line 347, please report the results in mm
Author Response
We would like to thank the Editor and the Reviewers of Plants MDPI for appreciation of our work. Corrections have been made to the manuscript according to reviewers’ comments and they are highlighted in yellow in the manuscript document. In addition, all comments are answered one by one.
Reviewer #1
We would like to thank Reviewer #1 for his/her time and his/her comments.
Comment 1: Line 20, replace less effective with ‘’less pronounced’’
Response 1: Correction has been made in lines 19-20 of the revised manuscript based on the reviewer’s comment.
Comment 2: Line 102, table 1. Since the values presented in Table 1 are also included in the text above, then it is not necessary to keep table 1. The authors should consider deleting it.
Response 2: Table 1 has been deleted based on the reviewer’s suggestion. The corresponding corrections, after the table deletion, have also been made throughout the manuscript (lines 124, 135, 136 and 148 of the revised manuscript).
Comment 3: Table 2 presents the antibacterial activities of pure SEO and inclusion complexes measured as inhibition zones. The results are presented in cm, however, the results should be presented in mm to coincide with the published literature. Please make the unit conversion.
Response 3: The unit conversion has been made in the table and wherever else necessary throughout the manuscript, based on reviewer’s comment (lines 149, 151, 340, 343 and Table 1 of the revised manuscript).
Comment 4: Also for table 2, please clarify for the reader in the footnote what it means when there is no value presented (‘‘-‘’ no antimicrobial effect)
Response 4: Footnote was added based on the reviewer’s suggestion in Line 153 of the revised manuscript.
Comment 5: Line 338, please add the corresponding number for each bacterial species.
Response 5: The bacterial species numbers have been added in the lines 334 and 335 of the revised manuscript.
Comment 6: Line 347, please report the results in mm
Response 6: Corrections have been made regarding the unit conversion throughout the manuscript (please see the corresponding response 3).
Reviewer 2 Report
The present study focused on the preparation of Sage essential oil (SEO)/ β-cyclodextrin (β-CD) inclusion complexes (ICs), and the evaluation of their antimicrobial activity against Escherichia coli, Staphylococcus aureus and 16 Listeria monocytogenes. The results showed that β-CD effectively formed inclusion complexes with SEO in satisfactory yields. The antimicrobial activity of the SEO in prepared complexes with β-CD was exhibited against L. monocytogenes and S. aureus. Several suggestions for further improvement of the quality of this manuscript are as follows,
1. More important data should be added in the abstract, and the method described in abstract should be more concise.
2. The major chemical composition of Sage essential oil should be provided, in addition to α-thujone 31.25%, what about other components in Sage essential oil.
3. How about the stability of sage essential oil before and after β-cyclodextrin encapsulation ? such as thermal stability and bioactivity stability.
4. For the chemical characterization of ICs, only FT-IR was performed. It is recommended to add other chemical analysis, such as NMR analysis, GC-MS analysis, and SEM analysis.
5. For the encapsulation of SEO, are there any optimization for the concentrations of β-CD and SEO ? why 5000 mg of β-CD and 670 mg of SEO were used in the present study ?
Author Response
We would like to thank the Editor and the Reviewers of Plants MDPI for appreciation of our work. Corrections have been made to the manuscript according to reviewers’ comments and they are highlighted in yellow in the manuscript document. In addition, all comments are answered one by one.
Reviewer #2
We would like to thank Reviewer #2 for his/her time and his/her comments.
Comment 1: More important data should be added in the abstract, and the method described in abstract should be more concise.
Response 1: In the abstract, an overview of the most important data and methodology used is already described. More details were avoided in order not to have a long abstract and to draw the attention of the reader to the presented results in the manuscript.
Comment 2: The major chemical composition of Sage essential oil should be provided, in addition to α-thujone 31.25%, what about other components in Sage essential oil.
Response 2: Additional information about the composition of sage essential oil has been added in the manuscript (lines 274, 275 of the revised manuscript). The major components are mentioned.
Comment 3: How about the stability of sage essential oil before and after β-cyclodextrin encapsulation? such as thermal stability and bioactivity stability.
Response 3: The study mainly focused on the antibacterial activity of pure SEO and inclusion complexes of SEO with β-CD. So, in this case, the thermal stability was not investigated. The bioactivity was determined only in terms of antimicrobial activity in vitro.
Comment 4: For the chemical characterization of ICs, only FT-IR was performed. It is recommended to add other chemical analysis, such as NMR analysis, GC-MS analysis, and SEM analysis.
Response 4: Indeed, the aforementioned analyses are widely used for the characterization of the ICs and they can provide a plethora of information mainly regarding the interactions of SEO and CD. However, in the case of the present study, the full ICs characterization was not necessary to be performed with such a wide variety of analyses. FT-IR and quantitative characterization were performed in order to verify the formation of the ICs, that were later examined for their antibacterial activity. Since the ICs formation of SEO with CD has not been reported in the literature, further future research may also focus on the full characterization of such ICs.
Comment 5: For the encapsulation of SEO, are there any optimization for the concentrations of β-CD and SEO? why 5000 mg of β-CD and 670 mg of SEO were used in the present study?
Response 5: These concentrations are based on the iso-molecular ratio between the two materials tested (SEO and β-CD). They were selected based on literature data (as described in lines 287-290) and preliminary experiments.
Reviewer 3 Report
The work presented in this manuscript deals with the use of encapsulated, in β-CD, sage essential oil, in order to improve its use as a food, cosmetic, pharmaceutical added agent. The antimicrobial effect that these inclusion bodies exhibit, against three common food borne bacterial strains, is also well documented by the authors.
It is a preliminary work with significant interest, very well presented, that deals with food preservation methods mainly. According to that I do not believe that this manuscript should be considered for publication in Plants.
So my suggestion is that this work should be submitted in food science and technology scientific journals.
Author Response
We would like to thank the Editor and the Reviewers of Plants MDPI for appreciation of our work. Corrections have been made to the manuscript according to reviewers’ comments and they are highlighted in yellow in the manuscript document. In addition, all comments are answered one by one.
Reviewer #3
We would like to thank Reviewer #3 for his/her time and his/her comments.
Comment: The work presented in this manuscript deals with the use of encapsulated, in β-CD, sage essential oil, in order to improve its use as a food, cosmetic, pharmaceutical added agent. The antimicrobial effect that these inclusion bodies exhibit, against three common food borne bacterial strains, is also well documented by the authors.
It is a preliminary work with significant interest, very well presented, that deals with food preservation methods mainly. According to that I do not believe that this manuscript should be considered for publication in Plants.
So my suggestion is that this work should be submitted in food science and technology scientific journals.
Response: The content of the present work is in line with the special issue ‘’Nanotechnological or Innovative Formulation Approaches for Efficient Delivery of Plant Ingredients II’’ of Plants MDPI. Also, since no application of the ICs in food products is reported in the study, but the formation, characterization and antimicrobial activity of ICs, probably the scope is more appropriate for this journal’s special issue, than food science and technology scientific journals.
Round 2
Reviewer 2 Report
The revised manuscript could be accepted.
Reviewer 3 Report
I have no more comments.
I suggest to accept the manuscript for publication